# SELF-DIRECTED DISCOVERY: HOW LLMS EXPLORE AND OPTIMIZE CONFIGURABLE LLM SOLUTIONS

## ABSTRACT

LLM solutions are increasingly replacing specialized machine learning models across various industry domains. While offering simplicity and maintainability advantages, optimizing these workflows remains heavily dependent on expert-driven experimentation. In this paper we test off-the-shelf powerful LLMs for the task of automatically building and iteratively improving LLM-based solutions. We propose a configuration-driven framework which defines such workflows by specifying model parameters, prompt templates, and data transformations. We show that this standardized representation enables automatic iterative improvement loops via optimization agents which are themselves defined within the same framework. When evaluated on challenging datasets, it discovers improved solutions while maintaining interpretability and human verifiability. The improvement loop we instantiate is generic and minimal (using prompts that have not been engineered) and self-referential (improved solutions are discovered with improved documentation and examples). The proposal is a self-improving system that bridges the gap between generic code-generating automatic optimization and more narrowly-focused techniques such as prompt engineering.

## 1 INTRODUCTION

Applications across domains such as financial services, healthcare, customer support, content moderation, and many others, increasingly rely on LLM solutions where the same inference pattern is applied to thousands or millions of data points. Such unified LLM-based solutions are increasingly preferred over specialized ML solutions, offering the appeal of a single, versatile approach that can handle diverse tasks without requiring domain-specific model training and maintenance (Chkirbene et al., 2024; Raza et al., 2025; Saleh et al., 2025). While these LLM inference workflows may initially lack the efficiency of highly optimized, custom machine learning (ML) solutions developed over years, maintainable systems often outweigh the performance gains of complex, specialized alternatives. However, unlike conversational AI systems where each interaction is unique, these LLM inference scenarios involve repeatedly executing identical prompt templates with only the input data varying, creating additional opportunities for optimization, for both quality and other performance goals. While it is becoming commonplace to utilize techniques such as prompt caching (Xiao et al., 2023; Zhu et al., 2023), automatic prompt engineering ((Ramnath et al., 2025; Zhou et al., 2022; Li et al., 2025a; Debnath et al., 2025)) or model selection (Wang et al., 2023; Tanaka et al., 2023; Tang et al., 2024), optimizing repeated inference patterns *holistically* still heavily relies on expert-driven experimentation (Li et al., 2024).

On the other hand, generative models are also transforming ML *research* by acting as powerful assistants that automate routine tasks, augment human creativity, and accelerate the pace of discovery. While LLMs are not yet capable of fully replacing human ML scientists, they significantly enhance efficiency by automating coding, data analysis, hypothesis generation, and manuscript writing (Tang et al., 2025; Wang & et al., 2025; Yang et al., 2024b; Schramowski et al., 2025). In this paper we take a step further in the direction of using generative models to aid in the development of ML solutions by showing that LLMs can successfully drive the optimization of LLM inference workflows and help discover improved solutions. This proposal addresses the gap between generic code-generating automatic optimization and more narrowly-focused techniques such as prompt engineering.

The paper makes the following contributions:

- We address a specific class of ML solutions based on LLM inference and propose a configuration-driven framework that supports *interpretability*. We demonstrate that this constrained yet expressive solution space enables powerful off-the-shelf LLMs to autonomously create and improve workflows across various tasks with minimal task-specific guidance. The generated workflows execute without errors over 95% of the time, achieve task performance that matches standardized prompts on average, and consistently discover significantly better solutions.

- We further implement a simple automatic iterative improvement loop, where other LLMs improve the solutions for custom goals such efficiency or quality. We propose a very simple linear iterative improvement loop using an Analyzer and an Improver agent, and show it can find better solutions to various challenging data sets. Since the Analyzer and Improver are in themselves LLM-based, we define them within the same framework and can thus be the object of optimization as well.

- We prioritize documentation-driven optimization over prompt engineering, which requires extensive manual tuning for each task and optimization objective. Instead of engineered meta-prompts, our LLMs receive minimal generic prompts alongside rich contextual artifacts: framework documentation and workflow examples. This approach enables robust cross-task generalization and sets the stage for a self-evolving system where accumulated knowledge artifacts improve performance across diverse domains without task-specific prompt crafting.

## 2 CONFIGURATION-DRIVEN LLM AGENT DEFINITION

In this section we test whether off-the-shelf LLMs can generate high-performing agents for new tasks given a well-defined agent specification framework and examples of agents implemented using it. To achieve this we propose a configuration-driven LLM agent framework which promotes interpretability (a human user can easily understand and verify the generated agent) and enables systematic optimization (another LLM can iteratively improve on these agents).

**LLM Agent**  We define an LLM agent as a workflow that makes a *single* LLM inference call, optionally preceded by input pre-processing and followed by output processing. [1]. More precisely, let $\mathcal{D} = \{(x_1, y_1), \ldots, (x_n, y_n)\}$ denote a dataset where each data point consists of input $x_i$ and optional ground truth $y_i$. An LLM agent $A$ is defined by:

- A prompt template $T$ containing fixed text and placeholder variables
- A set of input functions $\{f_1, f_2, \ldots, f_k\}$ that transform input data $x_i$
- An LLM model ($\theta_{\text{LLM}}$) alongside inference hyper-parameters $\rho$ (temperature, max tokens, etc.)

For input $x_i$, the agent generates a prompt $p_i$ and then output $o_i$ as :

$$p_i = T(f_1(x_i, c), \ldots, f_k(x_i, c), c) \quad \text{and} \quad o_i = A(x_i) = \theta_{\text{LLM}}(p_i; \rho)$$

where $c$ stands for additional context (examples, external data sources, etc.). Dataset-level performance is computed as $\bar{E} = \frac{1}{n} \sum_{i=1}^{n} E(o_i, x_i, y_i)$ where $E$ is a task-appropriate evaluation function.

**Agent Implementation**  At the level of implementation, we describe an LLM agent via three files: Configuration (json), Agent Class Implementation (Python) and Input Schema File (json). See Appendix B for more details. The *configuration* file specifies the agent's behavior: model settings, prompt template, input processing functions, and output format. The agent class implements the pre-processing functions referenced in the configuration. These functions handle data transformations, feature extraction, example retrieval, and other per-datapoint operations. The input schema file defines the agent's interface with other components, specifying what data the agent can access and how it communicates with upstream and downstream tasks. This three-file structure enforces clear separation between declarative configuration, implementation logic, and interface contracts, enabling systematic optimization while maintaining interpretability.

At run-time, agents process data points sequentially through a standardized pipeline: load and validate configuration files, apply input transformations, render the prompt template, call the LLM,

---

[1]This definition differs from the broader usage of "agent" in the literature, which typically encompasses multi-step reasoning, tool usage, memory systems, and iterative planning. Our constrained definition focuses on the fundamental building block of LLM-based solutions.

and parse the response according to the specified output format. Each agent is constrained to exactly one LLM inference call—multi-step reasoning requiring multiple LLM interactions must be orchestrated at the workflow level using separate agents.

**Modifiability Control** To guide other LLMs in improving task agents, we introduce modifiability control. Each configuration section includes a flag $modifiable \in$ true, false that helps meta-agents understand which sections can be modified. For example, model.modifiable $=$ true allows modification of model parameters, while output.modifiable $=$ false preserves the response schema. This enables systematic exploration while maintaining configurable constraints based on the specific improvement loop requirements—for example, preserving output schemas for downstream compatibility allowing flexible optimization across all components.

Appendix C shows a complete math problem solving agent example and Appendix D shows how this design supports diverse implementations including ML integration, in-context learning, and dynamic input transformation.

## 2.1 LLM-BASED AGENT GENERATION

To evaluate the configuration-driven framework, we first examine whether LLMs can generate valid task agents $A_t$ given only framework documentation and minimal task context. This evaluation aims to validate the framework's clarity and usability, as successful agent generation from documentation and minimal context indicates that the abstractions are sufficiently well-designed for automated solution synthesis.

### 2.1.1 EXPERIMENTAL SETUP

We evaluate agent generation capabilities across the following problems:

**AIME2024 and AIME 2025** contain 30 competition-level mathematics problems each, requiring integer answers between 000-999 (MAA Committees). These challenging problems span algebra, geometry, number theory, and combinatorics, requiring complex mathematical reasoning that remains difficult for current (non-reasoning) models. We evaluate exact match accuracy and use AIME2024 Part I for development, with remaining datasets for testing.

**Math500:** 500 competition mathematics problems with detailed solutions across 7 subjects (Algebra, Counting & Probability, Geometry, Intermediate Algebra, Number Theory, Precalculus, Prealgebra) (Hendrycks et al., 2021; Lightman et al., 2024). For Math500, evaluation uses mathematical expression normalization to account for equivalent mathematical representations[2]. We sample 100 data points for development and leave the remaining 400 for test.

**MMLU-Pro** is a benchmark designed to challenge LLMs beyond the original MMLU (Wang et al., 2024). It features reasoning-focused multiple-choice questions across 14 domains with ten answer options instead of four, creating a more demanding evaluation. We use validation data for development and sample 200 test problems.

We test if LLMs can generate working agents for these tasks using agent framework documentation and minimal task information. Specifically we implement the agent generation using a single meta-agent defined in the same framework. Table 1 shows the prompt template used by this meta-agent, where the placeholders stand for:

1. ⟨Framework Documentation⟩ and ⟨Agent Example⟩: The complete LLM Agent README (in supplementary material, totaling 5k tokens) and a single complete agent example. For all tasks we use the basic AIME2024 agent shown in Appendix C. Note that this example is out-of-domain for MMLU-Pro.

2. ⟨Task Data Point⟩ and ⟨Task description⟩: Task data points are randomly drawn from a different data split, with the purpose of exemplifying the task to be solved and the input/output format. Task descriptions are 1-2 sentences describing the task objective (see Table 1).

---

[2]We use the grader at `https://github.com/openai/prm800k/blob/main/prm800k/grading/grader.py`

Table 1: Prompt template used by the agent generating (meta-)agent, which generates task agents $A_t$ using the task information in this table, alongside the ⟨Framework Documentation⟩ and ⟨Agent Example⟩ described above.

| Dataset | ⟨Task Data Point⟩ | ⟨Task Description⟩ | Generator Prompt Template |
|---|---|---|---|
| AIME2024 | Every morning Aya goes for a 9-kilometer-long walk and stops at a coffee shop afterwards. When [...] Find the number of minutes the walk takes her, including the $t$ minutes spent in the coffee shop. Answer: 204 Solution: *detailed solution* | AIME (American Invitational Mathematics Examination) problems require integer answers between 000 and 999. The agent must show detailed reasoning and provide the final numerical answer. | Instructions: You are an expert AI agent developer. Task: ⟨Task Description⟩ Requirements: Generate exactly 3 files: (1) Agent configuration JSON, (2) Agent implementation Python, (3) Input schema JSON Schema. The agent must accept problem data as input, generate detailed reasoning, output correct format, handle parsing errors gracefully, and follow framework patterns. Framework Doc: ⟨Framework Documentation⟩ Agent Example: ⟨Agent Example⟩ Sample Data: ⟨Task Data Point⟩ Output Format: JSON only with agent_config, agent_code, input_schema fields. Generate the agent now: |
| AIME2025 | Find the sum of all integer bases $b > 9$ for which $17_b$ is a divisor of $97_b$. Answer: 070 | | |
| Math500 | Simplify $\tan 100° + 4\sin 100°$. Answer: $-\sqrt{3}$ Subject: Precalculus, Level: 2 | MATH problems require answers in LaTeX format within \boxed tags. The agent must show detailed reasoning and provide the final answer in the correct format. | |
| MMLU-Pro | Which of the following represents an accurate statement concerning arthropods? A. They possess an exoskeleton composed primarily of peptidoglycan. B. They possess an open circulatory [...] C. [...] Answer: B | MMLU-Pro problems are multiple-choice questions across academic subjects (biology, math, physics, etc.) with 10 answer choices (A-J). The agent must provide detailed step-by-step reasoning and select the correct answer choice. | |

The LLM must infer an appropriate agent structure, input processing logic, prompt design, and output schema solely from the framework documentation, complete agent example and data sample; no additional examples of agent configurations, code samples, or task-specific guidance are provided. However in the future we see this as a self-evolving framework, where the amount of artifacts increases with every problem solved, creating better and richer context for the LLM or code assistant.

**Evaluation and baselines**  In order to facilitate comparisons, we standardize Claude 3.5 Sonnet v2 for all task agents by removing other model references from the framework documentation. The meta-agent uses the same model with temperature 0.7 for exploration.

We run the meta-agent 10 times per task and perform static validation on generated configurations (structure, model names, required fields). Runtime issues include undefined placeholders (non-fatal) and syntax/execution errors (fatal). Success rate measures the percentage of agents executing without fatal errors on development data. For successful agents, we further evaluate task performance using multiple runs due to output variability: 5 runs for AIME datasets and 2 runs for others. We report accuracy averaged across all runs (Pass@1).

We compare against standard prompts for each task: basic CoT for AIME datasets, CoT with one example for Math500, and 5-shot CoT for MMLU-Pro. Exact prompts and evaluation methodology follow `https://www.vals.ai/benchmarks/`.

While the framework supports building sophisticated agents beyond prompt variations, we expect limited use of its full expressive power in these experiments given the minimal framework documentation and single agent example provided.

### 2.1.2 RESULTS

Results are shown in Table 2.

The meta-agent successfully generated executable agents across all datasets with 100% success rate, even for MMLU-Pro—a previously unseen task provided with only a single example and brief description. The generated agents demonstrated strong performance with median performance approaching that of the standard prompts for the tasks. On all tasks, the best agents significantly out-

Table 2: Performance when generating 10 agents for each task: Success rates (percentage of generated agents that execute without errors) and task performance metrics (median and max). Base accuracy reports performance of standard prompts used for these tasks (see `https://www.vals.ai/`)

| Dataset | Success Rate | Base Acc. | 1 Task data point | | 5 Task data points | |
|---|---|---|---|---|---|---|
| | | | Median Acc. | Max Acc. | Median Acc. | Max Acc. |
| AIME24 I | 100% | 22.7 | 21.3 | **25.3** | 20.7 | **25.3** |
| Math500 | 100% | 70.0 | 66.0 | **74.0** | 69.0 | 73.0 |
| MMLU-Pro | 100% | 82.0 | 82.1 | 87.1 | 82.9 | **90.0** |

performed the baselines, reaching maximum accuracies of 25% on AIME2024, 74% on Math500, and 90% on MMLU-Pro (an 8% absolute improvement). The primary challenge for agent generation was ensuring consistency between the output format specification in the configuration, the formatting instructions in the prompt template, and the evaluation requirements (which expect the same output fields as shown in the task examples). Misalignment across these components resulted in task agents that produced un-parseable outputs, leading to evaluation failures and zero accuracy scores.

All agents implemented dataset-specific preprocessing and incorporated detailed chain-of-thought reasoning through structured step-by-step problem breakdown. The AIME2024 best agent wrote a 6-step processes: "1. Read and understand the problem carefully, 2. Break down key information, 3. Plan your solution [...] 6. Verify answer is integer between 000-999" . In term of temperature. 80% of agents used temperature 0.7, while the rest used used 0.2 or 0.3. Input/output token patterns shows MMLU-Pro's best agent used more tokens than the average (+13% input, +17% output), while the other tasks showed minimal token differences. No agent incorporated worked examples or few-shot demonstrations. Finally, the agent generation performance remains similar across datasets regardless of whether 1 or 5 task examples are provided, demonstrating that models can understand new tasks from a single example.

## 3 DATA-DRIVEN AGENT IMPROVEMENT

We next explore whether LLMs can not only create agents but also improve them over time through iterative optimization, using meta-agents to analyze performance and generate better configurations.

We implement a simple self-improvement orchestration pattern involving four agents: a main task agent $A_{task}$ that is the target of improvement, evaluator $A_{eval}$, analyzer $A_{analyze}$, and improver $A_{improve}$. We employ simple scorers as $A_{eval}$ rather than LLM-based ones to ensure reliable performance measurements. Although these experiments focus on optimizing $A_{task}$, the analyzer and improver agents themselves can serve as optimization targets in the same iterative process.

More precisely, given a development dataset $\mathcal{D} = \{(x_i, y_i)\}_{i=1}^N$ and performance metric $\rho$, we seek to find agents $A'_{task}$ that improve $\Sigma_i \rho(A'_{task}(x_i), y_i)$.

**Improvement Loop:** At iteration $t$, the system executes:

$$\text{Out}_t = A_{task}^{t-1}(\mathcal{D}, \text{Ctx}) \tag{1}$$

$$\text{Eval}_t = A_{eval}(\text{Out}_t, \mathcal{D}) \tag{2}$$

$$\text{A}_t = A_{analyze}(\text{Eval}_t, \text{Out}_t, \mathcal{D}) \tag{3}$$

$$\text{A}_{task}^t = A_{improve}(\text{A}_{task}^{t-1}, \text{A}_{task}^{best}, \text{A}_t, \mathcal{H}_{t-1}, \text{Ctx}) \tag{4}$$

$$\mathcal{H}_t = \mathcal{H}_{t-1}; (\text{Out}_t, \text{Eval}_t, \text{A}_t) \tag{5}$$

where $\text{Out}_t$ are task outputs, $\text{Eval}_t$ contains evaluation metrics (aggregate and individual results), $\text{A}_t$ provides analytical insights, and $\mathcal{H}_t$ is the improvement history.

The analyzer's role it to identify patterns in performance and provide qualitative insights; we define an analyzer that uses aggregate results and randomly samples correct and incorrect examples from the data. The improver uses current and best task agents, current performance analysis and historical

data, and is instructed to write a new agent while respecting the *modifiable* flag. The process starts with an initial agent $A_{\text{task}}^0$, and terminates when a target performance is met or maximum number of iterations is reached. With respect to the optimization objective, the loop implements metric-agnostic improvement by embedding the target metric within the evaluator, allowing the meta-agents to implicitly understand and optimize toward the desired goal.

The Analyzer and Improver agents are given in Appendix E and F respectively, while their prompt templates are summarized in Appendix I. Importantly, as it can be observed, these agents have minimal prompts and rely on the documentation artifacts that are provided to them (Ctx): the README associated to the LLM framework (as in the previous section) and an additional README file describing the iterative improvement loop.

The Analyzer and Improver agents are detailed in Appendices E and F, with prompt templates in Appendix I. These meta-agents use minimal prompts and instead rely on documentation (Ctx): the LLM framework README and an additional README describing the iterative improvement process. This is an artifact-driven approach where rich contextual information replaces engineered prompts.

## 3.1 EXPERIMENTS

We included math tasks in our experiments because they remain challenging for pre-reasoning state-of-the-art models and exhibit test-time scaling where longer outputs improve accuracy (Snell et al., 2024; Muennighoff et al., 2025). This property creates a natural optimization space: agents can achieve better quality through longer reasoning chains at higher computational cost, or maintain quality while reducing token usage, making them ideal testbeds for iterative improvement loops.

This section introduces additional pseudo-random number generator (PRNG) datasets specifically designed to test test-time scaling properties. Unlike other tasks where models perform pattern recognition, PRNG sequence prediction offers no shortcuts and requires precise algorithmic execution. We use three PRNG algorithms (BBS/LCG/MWC) with deliberately small constants to be more approachable for LLMs, which do not excel at large number arithmetic. We set the task of predicting the $k$-th number in the sequence with $k = 2$, based on the observation that even strong models perform poorly on this task. In contrast, reasoning models (GPT-OSS-120b) generate long reasoning traces and achieve 100% accuracy even for larger $k$ values. We implement an initial PRNG solver agent as a generic CoT code execution agent. See Appendix G for details on data set creation and PRNG agent.

### 3.1.1 ACCURACY OPTIMIZATION

We conduct 10-iteration improvement cycles using the same Analyzer/Improver across all tasks. We test Sonnet 3.5 v2 and GPT-OSS-120b as meta-agent LLMs and restrict task models to Sonnet 3.5 v2. As in previous sections, we use an Evaluator that computes accuracy and provides aggregate and per-problem scores.

Table 3 shows results on development and test splits. The "Original agent" uses the standard prompt from the previous section and serves as the starting point for all improvement loops. The "Best agent" column shows the highest-performing configuration from the previous section's agent generation experiments (performed on the development set).

**AIME and MMLU-Pro** All agents achieve improved performance on development sets. The challenging AIME datasets show modest improvements ($0.22 \rightarrow 0.27$ with Sonnet 3.5 v2). MMLU-Pro demonstrates strong overall performance, with substantial development improvements ($0.82 \rightarrow 0.89/0.90$) translating to more modest but consistent test gains. Overall, the results indicate that only larger development improvements transfer to test gains, suggesting insufficient development data or that baseline configurations had already undergone significant optimization in prior work. In terms of solutions found, the best AIME agent identifies problem types using word matching and provides targeted guidance such as "Count systematically by cases" for grid problems and "Factor completely" for number theory problems.

Appendix H shows a typical analyzer output from Sonnet analyzers. On AIME for example, analyzers commonly: identify where agents make arithmetic errors or use incorrect formulas, observe performance across problem types (e.g. low on geometry), highlight cases where multiple solu-

Table 3: Performance of initial configurations and after improvement loops. Original agent is the standard task prompt and serves as the $A_{\text{task}}^0$. Best agent reports on the best configuration detected in the previous section.

| Dataset | Original Agent | Best Agent | Improvement Loop | |
| --- | --- | --- | --- | --- |
| | | | Sonnet 3.5 v2 | GPT-OSS-120b |
| AIME2024 I (dev) | 0.22 | 0.25 | **0.27** | 0.25 |
| AIME2024 II | 0.07 | 0.05 | **0.11** | 0.03 |
| AIME2025 | **0.05** | 0.03 | 0.03 | 0.03 |
| MMLU-Pro-dev | 0.82 | **0.90** | 0.89 | 0.84 |
| MMLU-Pro | 0.79 | **0.83** | 0.80 | 0.77 |
| PRNG-bbs | 0.21 | - | 0.57 | **1.0** |
| PRNG-lcg | 0.0 | - | 0.02 | **1.0** |
| PRNG-mwc | 0.0 | - | 1.0 | **1.0** |

tions yield different answers (and recommend lower temperature), reason about the optimization trajectory, or highlight the need for more exploration by increasing temperature. Improvement loops (plotted in Appendix K) are not monotonic, reflecting that the optimization process is mostly exploratory, with agents often showing temporary performance degradation before discovering more effective approaches.

**PRNG tasks** For PRNG tasks, the two meta-agents discovered very different solutions. By iterations 3 or 4, GPT-OSS-120b invariantly discovered agents that pre-compute correct answers and instruct the LLM to output them. Specifically, the Analyzer, which sees correct/incorrect examples, inferred that all inputs list the same algorithm and instructed the Improver to create agents that execute this algorithm. The exact implementation varied: for example inserting a prompt section "verification_note" which instructs the LLM to check the answer against the pre-computed answer or a section "generate_example_trace" which despite the name contains the actual computed execution trace for that data point. While not technically cheating, these solution assume that the observed examples are representative of the true data distribution. In contrast the Sonnet loops only discovered this solution for the MWC algorithm and failed to find a working solution for the LCG algorithm. For BBS, Sonnet 3.5 finds improved solutions which list explicit modulo arithmetic rules (which the Analyzer identifies as difficult), add examples and clear validation guidance, but do not compute the function deterministically (reaching 57% performance). Regarding test-time scaling, the system did not identify it as a reliable performance improvement strategy. While one improvement loop produced agents that generated 50% more output tokens (with 10% longer input), this increased verbosity did not consistently translate to better performance. Appendix J shows an extreme case where detailed reasoning instructions resulted in outputs twice the length of the starting agent.

**Variation across loops** We observe that results vary when running identical improvement loops multiple times. To explore this, we run each Sonnet loop 4 additional times and observe max scores in the range [0.84-0.89] for MMLU-Pro, [0.24-0.28] for AIME, and [0.57-0.93] for PRNG-bbs. Combined with the observation that larger development improvements generalize better, this suggests the improvement loop in its current form serves as a discovery tool rather than monotonic optimization. Despite this variability, a *single* 10-iteration loop discovers superior solutions within an effectively infinite search space of pre-processing functions, parameters, and prompt templates, showing very good optimization efficiency.

### 3.1.2 ACCURACY+COST OPTIMIZATION

We next explore optimization under different objectives using more complex initial configurations. To balance accuracy and efficiency, we introduce a cost-accuracy evaluator where cost equals input_tokens + 3 × output_tokens (reflecting real-world pricing). The scoring function assigns 0 points for incorrect answers and 1/cost for correct ones, prioritizing accuracy while minimizing cost as a secondary objective. We modify optimization goals by changing the evaluator's scoring computation and description. Figure 1 shows fragments of an evaluator output.

```
1    "overall_accuracy": 0.267,
2    "overall_score": 2.1e-05,
3    "avg_input_tokens": 13006.1,
4    ...
5    "optimization_goal": "Minimize cost while maintaining accuracy",
6    "scoring_explanation": "Score: 0 if wrong answer, 1/cost if [...]"
```

Figure 1: Example evaluator output for cost+accuracy optimization. The explanation fields are intended to guide the optimization goals of the meta-agents
.

We experiment on AIME and MMLU-Pro datasets. MMLU-Pro uses the CoT+ICL agent from the previous section as $A^0_{\text{task}}$. For AIME experiments, we start with an ICL agent that samples 2 examples per problem from the s1k dataset (Muennighoff et al., 2025), which contains similar problems with reasoning traces from powerful models. This dataset has been shown to improve AIME performance when used to fine-tune models.

Table 4: Performance of initial configurations and after improvement loops for cost+accuracy optimization: Accuracy and average input/output tokens.

| Dataset | Original Config | | | Improvement Loop Sonnet 3.5 v2 | | | Improvement Loop GPT-OSS-120b | | |
|---|---|---|---|---|---|---|---|---|---|
| | Acc | In | Out | Acc | In | Out | Acc | In | Out |
| AIME2024 I (dev) | 0.24 | 12.8k | 340 | 0.33 | 6.1k | 347 | 0.29 | 478 | 197 |
| AIME2024 II | 0.09 | 13.9k | 368 | 0.08 | 5.9k | 350 | 0.07 | 483 | 209 |
| AIME2025 | 0.05 | 14.2k | 365 | 0.05 | 6.0k | 334 | 0.03 | 488 | 216 |
| MMLU-Pro (dev) | 0.82 | 760 | 190 | 0.84 | 301 | 176 | 0.84 | 230 | 196 |
| MMLU-Pro | 0.79 | 805 | 192 | 0.78 | 256 | 172 | 0.79 | 275 | 196 |

Results are shown in Table 4. Both Sonnet and GPT models explored much more diverse solutions than in previous experiments, likely due to the more complex starting agent (for AIME) and the ambiguous cost function allowing both accuracy and cost optimizations. For AIME, agents implemented strategies including: 1) automatic problem categorization into geometric, algebraic, and combinatorial types, 2) targeted prompts for each problem type (e.g., for geometry: "1. Draw and label key elements, 2. List given measurements and relationships..."), and 3) type-based ICL example selection. This approach yielded the winning Sonnet solution. However, the winning GPT-OSS-120b agent (Appendix L) used none of these techniques, instead replacing random ICL sampling with a fixed geometry example. While cost-effective, this did not generalize to good test performance. The variation in discovered solutions suggests the development set may be too small for reliable AIME optimization. On MMLU-Pro, all agents were able to find much more efficient solutions in terms of cost with roughly 70% reduction in input tokens and no impact on test performance.

Overall the cost+accuracy optimization showed the potential for creative solution discovery, with agents autonomously developing problem classification systems, targeted prompting strategies, and token compression techniques. More sophisticated improvement patterns could incorporate larger development sets, extended exploration phases, or multi-objective optimization strategies in order to further expand the range of discoverable solutions.

## 4 RELATED WORK

**Large Language Models as Optimizers** LLMs have emerged as powerful optimization tools across diverse domains. Automatic prompt engineering (Ramnath et al., 2025; Zhou et al., 2022; Li et al., 2025a; Debnath et al., 2025) has demonstrated LLMs' ability to generate and refine prompts, with APE Zhou et al. (2022) introducing instructions as *programs* that can be systematically optimized. Similar work addresses automatic model selection (Wang et al., 2023; Tanaka et al., 2023; Tang et al., 2024), ICL example selection (Liu et al., 2022; Zhang et al., 2022; Do et al., 2024),

or prompt compression (Jiang et al., 2023; Li et al., 2025b) among many others. Recent advances explore LLMs as general-purpose optimizers (Jiang et al., 2025a; Yang et al., 2024a; Fernando et al., 2023), leveraging natural language understanding to interpret problems, generate solutions, and analyze feedback iteratively.

Building on these, our work introduces workflow optimization that extends beyond individual components to optimize entire agent configurations simultaneously: the system shows solution discovery capabilities that go beyond prompt variations. The approach enables metric-agnostic optimization through modular evaluator components, supporting flexible natural language-expressed optimization goals. Additionally, the self-referential optimization where meta-agents are defined within the same configurable framework as the task agents they optimize, enables the improvement process to enhance itself. Finally we designed this as an artifact-driven framework which leverages rich contextual documentation and examples to create a self-evolving system that accumulates knowledge rather than requiring extensive meta-prompt engineering.

**Autonomous Agents and LLM Workflow Optimization Frameworks**  Research in LLM-based systems has advanced along two complementary but very related directions: autonomous agents for dynamic problem-solving and optimization frameworks for systematic workflow improvement.

Autonomous Agent Systems have demonstrated remarkable capabilities in dynamic environments. ReAct Yao et al. (2023) integrates reasoning with external tools through "thought-action-observation" loops, enabling LLMs to reason, perform actions, and learn from outcomes. PAL Gao et al. (2022) focuses on programmatic reasoning by generating executable code as intermediate steps, e.g. offloading complex calculations to interpreters for improved accuracy on mathematical tasks. AutoGPT and similar frameworks create agents that independently set goals, break down tasks, and execute multi-step plans with minimal human intervention. AIDE Jiang et al. (2025b) introduces tree-search strategies for systematic code and ML pipeline optimization, structuring solutions hierarchically for targeted improvements based on incremental changes.

Many optimization frameworks have emerged to systematically improve LLM workflows. Some examples are DSPy (Khattab et al., 2023), which enables "programming, not prompting" through modular pipeline definition and automatic prompt optimization. TextGrad (Huang et al., 2023) introduces "automatic differentiation via text," using LLM-based feedback for iterative refinement within computational graphs. GPTSwarm (Zhuge et al., 2024) represents agents as optimizable computational graphs, unifying prompt engineering techniques through node-level optimization and edge-level orchestration. Teola Zheng et al. (2024) provides end-to-end optimization through dataflow graph representations, enabling rule-based optimizations across workflow modules.

Our approach bridges these paradigms through simple primitives that enable structured, interpretable optimization that combines the systematic exploration of optimization frameworks with the adaptive improvement mechanisms of autonomous agents. Unlike open-ended autonomous exploration that can produce black-box improvements, the configuration-driven approach ensures all generated solutions remain human-verifiable. Instead of replacing humans, the system can serve as a solution discovery tool that aids human decision-making through transparent improvement trajectories. This is possible through the constrained but expressive solution space, resulting in a middle ground between rigid optimization frameworks and unconstrained exploration systems.

## 5 Future Work

Current improvement cycles analyze development data point-by-point. While necessary for runtime execution, development stages could benefit from global dataset processing to enable aggregate analysis and dataset-wide optimization strategies not visible in individual examples. Additionally, more sophisticated improvement patterns could incorporate multi-objective optimization or Monte Carlo solution search, and beyond fixed meta-agents, the configuration-driven design provides a foundation for joint optimization of both task and meta-agents.

Finally, the framework can be extended to support cross-domain knowledge transfer by accumulating improvement artifacts across domains. This would build a repository of successful patterns and strategies, with accumulated knowledge informing improvements on new tasks and enabling transfer learning at the agent architecture level.

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

## A  GENERATIVE AI USE

AI tools were used throughout this research to enhance productivity and improve the quality of the work.

**Framework Development Process**  The framework and the framework documentation was developed iteratively with assistance from a code assistant powered by Sonnet 3.5 and Sonnet 4.0, depending on availability. The assistant was given specific instructions to improve documentation clarity and update the framework based on feedback. This iterative process was validated by testing the framework on toy prediction tasks (separate from the experimental tasks) to ensure the generated agents functioned correctly.

**Paper writing**    LLMs assisted in refining the clarity and coherence of the writing, suggesting more concise phrasings and identifying areas where technical concepts could be explained more clearly for broader accessibility. Additionally, AI tools facilitated data analysis and presentation by generating LaTeX tables and figures from raw experimental results. All AI-generated content was carefully reviewed and validated by the authors to ensure accuracy and appropriateness for the scientific context.

# B  LLM AGENT IMPLEMENTATION

**LLM Agent Implementation**    At the level of implementation, we describe an LLM agent via three components: Configuration (json), Agent Class Implementation (Python) and Input Schema File.

The **configuration** file defines the agent's behavior, model settings, input processing, prompt structure, and output format. A configuration tuple $C$ consists of:

$$C = \langle \text{desc}, \text{schema}, \text{model}, \text{inputs}, \text{prompt}, \text{output}, \text{settings} \rangle$$

where:

- desc: Human-readable description of agent purpose
- input schema: JSON Schema path defining expected input data structure (including additional context data).
- model $= \langle \text{name}, \rho \rangle$: LLM specification ($\theta_{\text{LLM}}$ and inference parameters $\rho$)
- inputs $= \{f_1, \ldots, f_k\}$: Data transformation functions operating on $x_i$ and context $c$
- prompt $= \langle T, \text{sections} \rangle$: Template $T$ with reusable sections and placeholders
- output $= \langle \text{format}, \text{schema}, \text{errors} \rangle$: Expected output format which drives the LLM response parsing
- settings: Execution parameters such as timeouts

Notably, inputs can be of two types: "data" or "computed" (with function name and arguments), in which case the function needs to be defined in the agent implementation class. For example the following configuration fragment shows the input section and references *raw_problem_data* which needs to exist in the input data processed, and *problem_text* which is computed via a function and takes the raw data as input.

Listing 1: Configuration example showing computed inputs

```
"inputs": {
  "raw_problem_data": {
    "source": "data"
  }
  "problem_text": {
    "source": "computed",
    "function": "extract_problem_text",
    "args": {"problem_data": "raw_problem_data"}
  }
}
```

The role of the **agent class** is to implement the functions referenced in the configuration's computed inputs. These functions operate at data-point level and can apply custom pre-processing (such as normalization, feature extraction, problem type classification, etc), retrieve examples from static resources provided as context $c$, etc. In the example above, the python agent class needs to implement the *extract_problem_text* function, for example as:

```
def extract_problem_text(self, problem_data):
    """Function referenced in config's computed inputs"""
    return problem_data.get('problem', '')
```

The **input schema file** and the output section of the configuration define the communication with upstream and downstream tasks. The agent can chose to ignore parts of the input data but can't access any resources outside what is defined in the input schema file.

**Modifiability Control**  In order to simplify the process of other LLMs improving task agents, we introduce modifiability control. Each configuration section includes a flag $modifiable \in \{\text{true}, \text{false}\}$ that guides other agents wrt which sections can be modified. For example, model.modifiable $=$ true indicates that model name or model hyper-parameters can be modified, while output.modifiable $=$ false indicates that the response schema needs to be preserved. This enables systematic exploration

of the configuration space while maintaining structural invariants essential for downstream processing, as well as the option to configure preferences (for example to prevent changes in the underlying LLM model if so desired).

**Agent Execution** The agents operate synchronously, processing one input data point at a time in sequential order. The framework reads the agent definition from three files (JSON configuration, Python implementation, and input schema), performs static validation of the configuration and dynamic validation of runtime input data. At runtime, the framework processes input data through defined transformations, generates the final prompt by resolving placeholders, and calls the LLM with the rendered prompt. The LLM response is then parsed by an output parser according to the output format specified in the configuration.

## C  MATH PROBLEM SOLVING AGENT

Listing 2: AIME 2024 Task Agent Configuration

```
1  {
2    "description": "AIME 2024 task agent - generates detailed reasoning followed by 3-digit
          integer answer",
3    "input_schema": "agents/task/task_input_schema.json",
4    "model": {
5      "_modifiable": true,
6      "name": "anthropic.claude-3-5-sonnet-20241022-v2:0",
7      "parameters": {
8        "temperature": 0.7,
9        "max_tokens": 8192
10     }
11   },
12   "inputs": {
13     "_modifiable": true,
14     "problem_text": {
15       "source": "computed",
16       "function": "extract_problem_text",
17       "args": {"problem_data": "raw_problem_data"},
18       "description": "Extracted problem text from AIME2024 format"
19     },
20     "raw_problem_data": {
21       "source": "data",
22       "path": "$",
23       "required": true,
24       "description": "Complete problem data from AIME2024 dataset"
25     }
26   },
27   "prompt": {
28     "_modifiable": true,
29     "sections": {
30       "instructions": "You are a mathematical problem solver specializing in AIME problems.",
31       "task_description": "AIME problems require integer answers between 000 and 999. Show
            detailed step-by-step reasoning, then provide the final numerical answer.",
32       "output_format": "You must respond with a JSON object containing exactly two fields:\n-
            \"reasoning\": string containing your detailed step-by-step solution\n- \"answer\":
             string containing your 3-digit answer (pad with leading zeros if needed)"
33     },
34     "template": "### Instructions:\n{prompt.sections.instructions}\n\n### Task:\n{prompt.
          sections.task_description}\n\n### Problem:\n{inputs.problem_text}\n\n### Output
          Format:\n{prompt.sections.output_format}\n\nNow solve the problem and respond with
          the JSON object:"
35   },
36   "output": {
37     "_modifiable": false,
38     "format": "json",
39     "schema": {
40       "type": "object",
41       "required": ["reasoning", "answer"],
42       "properties": {
43         "reasoning": {"type": "string", "description": "Detailed step-by-step solution"},
44         "answer": {"type": "string", "pattern": "^[0-9]{3}$", "description": "3-digit integer
              answer (000-999)"}
45       }
46     },
47     "error_values": {
48       "reasoning": "Error occurred during reasoning.",
49       "answer": "PARSE_ERROR"
50     }
51   },
52   "settings": {
53     "_modifiable": false,
54     "class_name": "AIME2024TaskAgent",
55     "timeout_seconds": 300
56   }
57 }
```

Listing 3: AIME 2024 Task Agent Implementation

```python
1  from llm_agent import LLMAgent
2  from typing import Dict, Any
3
4  class AIME2024TaskAgent(LLMAgent):
5      """
6      AIME2024-specific reasoning + answer agent.
7
```

```
    Processes AIME mathematical competition problems and generates detailed step-by-step
        reasoning followed by answers.
    """

    def __init__(self, name="AIME2024_Task", config_path=None, dry_run=False):
        super().__init__(name, config_path, dry_run)

    def extract_problem_text(self, problem_data: Dict[str, Any]) -> str:
        """
        Extract problem text from AIME2024 data format.

        AIME2024 format:
        {
            "id": int,
            "problem": str,  # Main problem statement
            "url": str
        }
        """
        return str(problem_data.get('problem', ''))
```

Listing 4: AIME 2024 Task Agent Input Schema

```json
{
  "type": "object",
  "required": ["id", "problem"],
  "properties": {
    "id": {
      "type": "integer",
      "description": "Problem identifier"
    },
    "problem": {
      "type": "string",
      "description": "The AIME problem statement"
    },
    "url": {
      "type": "string",
      "description": "URL to problem source (optional)"
    }
  }
}
```

## D  FRAMEWORK CAPABILITIES

The LLM Framework provides extensive flexibility for implementing diverse agent architectures and optimization strategies through its configuration-driven design. The framework's expressive power enables sophisticated agent implementations across multiple dimensions:

**Feature Extraction and ML Integration:** Agents can incorporate external machine learning models and feature extraction pipelines through initialization methods. Complex preprocessing can be implemented in the agent's __init__ method, loading pre-trained models, statistical analyzers, or domain-specific feature extractors. External datasets for feature computation can be specified in the configuration's input section and loaded during agent initialization:

```
"inputs": {
  "feature_dataset": {
    "source": "data",
    "path": "external_features.json",
    "description": "External dataset for feature extraction"
  },
  "extracted_features": {
    "source": "computed",
    "function": "extract_ml_features",
    "args": {"raw_data": "problem_data", "feature_db": "feature_dataset"}
  }
}
```

**In-Context Learning Implementation:** ICL can be most flexibly implemented as computed inputs that dynamically generate examples using various example selection strategies, including similarity-based retrieval, difficulty-matched sampling, or domain-specific filtering. ICL examples can be sourced from multiple datasets with automatic data leakage prevention:

```
"icl_examples": {
  "source": "computed",
  "function": "generate_icl_examples",
  "args": {
    "icl_data_path": "data/training_examples.json",
    "num_examples": 5,
    "similarity_metric": "cosine",
    "exclude_current": true
  }
}
```

**Dynamic Input Transformation:** The computed input system enables arbitrary feature engineering and input pre-processing. Agents can implement domain-specific transformations, adaptive input formatting, etc. Each transformation function can access multiple input sources and apply complex processing logic:

```
"processed_input": {
  "source": "computed",
  "function": "transform_input",
  "args": {
    "text_data": "raw_text",
    "numerical_data": "raw_numbers",
    "context_data": "external_context",
    "transformation_type": "hierarchical_fusion"
  }
}
```

**Model and Parameter Optimization:** The framework supports dynamic model selection and parameter tuning through the modifiable model configuration. Agents can be optimized across dif-

ferent LLM models with varying capabilities and costs. Model parameters including temperature, max_tokens, and model-specific settings can be systematically explored:

```
"model": {
  "_modifiable": true,
  "name": "anthropic.claude-3-5-sonnet-20241022-v2:0",
  "parameters": {
    "temperature": 0.7,
    "max_tokens": 4096,
    "top_p": 0.9
  }
}
```

**Prompt Engineering and Structure:** The modular prompt system enables sophisticated prompt architectures with independently optimizable sections. Agents can implement hierarchical prompting, conditional prompt sections, dynamic example integration, and task-specific reasoning templates. The template system supports complex placeholder substitution and structured prompt composition.

**Orchestration Constraints:** While the framework provides extensive flexibility, certain constraints ensure orchestration compatibility. The output schema should remain fixed (_modifiable: false) to maintain consistency across the evaluation-analysis-improvement loops. This constraint ensures that orchestration agents can reliably process results while allowing freedom in input processing, model selection, and reasoning strategies.

# E ANALYZER AGENT

## Listing 5: Analyzer Agent Configuration

```
1  {
2    "description": "Generic analyzer agent - analyzes task performance and identifies
          improvement opportunities",
3    "input_schema": "analyzer_input_schema.json",
4    "model": {
5      "_modifiable": true,
6      "name": "anthropic.claude-3-5-sonnet-20241022-v2:0",
7      "parameters": {
8        "temperature": 0.3,
9        "max_tokens": 8192
10     }
11   },
12   "inputs": {
13     "_modifiable": true,
14     "current_task_config": {
15       "source": "data",
16       "path": "current_task_config",
17       "required": true,
18       "description": "Current task agent configuration as JSON string"
19     },
20     "evaluation_summary": {
21       "source": "data",
22       "path": "evaluation_summary",
23       "required": true,
24       "description": "Aggregated evaluation metrics from evaluator"
25     },
26     "individual_results": {
27       "source": "data",
28       "path": "individual_results",
29       "required": true,
30       "description": "Per-problem results with model outputs, ground truth, and scores"
31     },
32     "correct_examples": {
33       "source": "computed",
34       "function": "extract_correct_examples",
35       "args": {
36         "individual_results": "individual_results",
37         "num_examples": 3,
38         "score_field": "score_field",
39         "perfect_score": "perfect_score"
40       },
41       "description": "Examples where the task agent performed correctly"
42     },
43     "incorrect_examples": {
44       "source": "computed",
45       "function": "extract_incorrect_examples",
46       "args": {
47         "individual_results": "individual_results",
48         "num_examples": 5,
49         "score_field": "score_field",
50         "perfect_score": "perfect_score"
51       },
52       "description": "Examples where the task agent made errors"
53     }
54   },
55   "prompt": {
56     "template": "### Instructions:\n{prompt.sections.instructions}\n\n### Current Task Agent
          Configuration:\n{inputs.current_task_config}\n\n### Evaluation Summary:\n{inputs.
          evaluation_summary}\n\n### Examples Where Task Agent Was CORRECT:\n{inputs.
          correct_examples}\n\n### Examples Where Task Agent Made ERRORS:\n{inputs.
          incorrect_examples}\n\nAnalyze the task agent performance and identify improvement
          opportunities:"
57   },
58   "output": {
59     "_modifiable": false,
60     "format": "json",
61     "schema": {
62       "type": "object",
63       "required": ["analysis"],
64       "properties": {
65         "analysis": {
66           "type": "string",
67           "description": "Complete analysis of task agent performance"
68         }
69       }
70     }
```

```
71      },
72      "settings": {
73        "_modifiable": false,
74        "class_name": "AnalyzerAgent",
75        "timeout_seconds": 600
76      }
77    }
```

Listing 6: Analyzer Agent Implementation

```python
1  from llm_agent import LLMAgent
2  from typing import Dict, List, Any
3  import json
4  import random
5
6  class AnalyzerAgent(LLMAgent):
7      """
8      Generic analyzer agent for performance analysis.
9
10     Analyzes task agent performance and identifies improvement opportunities
11     for any domain that follows the standard evaluator contract.
12     """
13
14     def __init__(self, name="Analyzer", config_path=None, dry_run=False):
15         super().__init__(name, config_path, dry_run)
16
17     def extract_correct_examples(self, individual_results: List[Dict[str, Any]],
18                                  num_examples: int,
19                                  score_field: str = "score",
20                                  perfect_score: float = 1.0) -> str:
21         """
22         Extract examples where the task agent performed correctly.
23         """
24         correct_examples = [result for result in individual_results
25                         if result.get(score_field, 0) >= perfect_score]
26
27         selected = random.sample(correct_examples, min(num_examples, len(correct_examples)))
28         return json.dumps(selected, indent=2)
29
30     def extract_incorrect_examples(self, individual_results: List[Dict[str, Any]],
31                                    num_examples: int,
32                                    score_field: str = "score",
33                                    perfect_score: float = 1.0) -> str:
34         """
35         Extract examples where the task agent made errors.
36         """
37         incorrect_examples = [result for result in individual_results
38                         if result.get(score_field, 0) < perfect_score]
39
40         selected = random.sample(incorrect_examples, min(num_examples, len(incorrect_examples)
41             ))
         return json.dumps(selected, indent=2)
```

# F Improver Agent

Listing 7: Improver Agent Configuration

```json
{
  "description": "Generic improver agent - generates improved task agent based on analysis",
  "input_schema": "improver_input_schema.json",
  "model": {
    "_modifiable": true,
    "name": "anthropic.claude-3-5-sonnet-20241022-v2:0",
    "parameters": {
      "temperature": 0.3,
      "max_tokens": 25000
    }
  },
  "inputs": {
    "_modifiable": true,
    "analysis_results": {
      "source": "data",
      "path": "analysis_results",
      "required": true,
      "description": "Analysis results from the analyzer agent"
    },
    "current_task_agent": {
      "source": "data",
      "path": "current_task_agent",
      "required": true,
      "description": "Current task agent configuration"
    },
    "best_task_agent": {
      "source": "data",
      "path": "best_task_agent",
      "required": false,
      "description": "Best performing task agent configuration and implementation"
    }
  },
  "prompt": {
    "template": "### Instructions:\n{prompt.sections.instructions}\n\n### Current Task Agent:\
        n{inputs.current_task_agent}\n\n### Best Performing Agent:\n{inputs.best_task_agent}\
        n\n### Analysis Results:\n{inputs.analysis_results}\n\nGenerate improved task agent
        configuration based on the analysis:"
  },
  "output": {
    "_modifiable": false,
    "format": "json",
    "schema": {
      "type": "object",
      "required": ["explanation", "new_task_implementation", "new_task_config"],
      "properties": {
        "explanation": {
          "type": "string",
          "description": "Detailed explanation of improvements"
        },
        "new_task_implementation": {
          "type": "string",
          "description": "Complete Python file content"
        },
        "new_task_config": {
          "type": "string",
          "description": "Complete JSON config content"
        }
      }
    }
  },
  "settings": {
    "_modifiable": false,
    "class_name": "ImproverAgent",
    "timeout_seconds": 1200
  }
}
```

Listing 8: Improver Agent Implementation

```python
from llm_agent import LLMAgent

class ImproverAgent(LLMAgent):
    """
    Generic improver agent - generates improved task agents based on analysis.
    """
```

```
 7
 8    def __init__(self, name="Improver", config_path=None, dry_run=False):
 9        super().__init__(name, config_path, dry_run)
```

# G PRNG EXPERIMENTS

We use the three algorithms BBS/LCG/MWC, and deliberately set small constants to be more approachable for LLMs which do not excel at large number arithmetic:

- **LCG**: $X_{n+1} = (25173 \cdot X_n + 13849) \bmod 65536$ - smaller modulus than typical implementations
- **MWC**: $X_{n+1} = (18782 \cdot X_n + \text{carry}) \& 0\text{xFFF}$ - 12-bit output with carry propagation
- **BBS**: $X_{n+1} = X_n^2 \bmod 209$ where $209 = 11 \times 19$ - small primes

We create train and test data sets (each containing 70 sequences) where models must predict the $k$'th number in the sequence given: (1) complete algorithm implementation in Python, (2) hyperparameters (e.g. the prime numbers used) and (3) initial seed value (distinct in train/test splits). We set the task as that of predicting the second number in the sequence ($k = 2$) based on the initial observation that even state-of-the-art models perform poorly on the task. In contrast, we confirm that a reasoning model (OpenAI 120b) generates long reasoning traces and achieves 100% accuracy.

Table 5: PRNG agents for BBS/LCG/MWC algorithms. The names of the algorithms are obfuscated in case the LLMs are aware of these algorithms and have memorized sequences. ⟨Function⟩ is the Python code, ⟨N⟩ is the number to be predicted (in this case at position 2), ⟨Initial Value⟩ the seed, ⟨Hyperparams⟩ are constants used in the algorithms. Note that this is a generic code execution agent, as all the placeholders are only resolved at run-time when individual data points are processed.

---

Instructions: You are a function evaluator. You will be given a function implementation, its hyperparameters, and an initial value. Your task is to predict what the function will return at a specific position. Task: Given the function implementation and parameters, determine what value the function would return at position ⟨N⟩ when called with the initial value. Note: The first number generated is at position 1, the first iteration is position 1. Show your detailed step-by-step reasoning, then provide the final prediction.

Function Implementation: ⟨Function⟩

Hyperparameters: ⟨Hyperparams⟩

Initial Value: ⟨Initial Value⟩

Approach: 1. Carefully analyze the function implementation 2. Understand what the hyperparameters control 3. Trace through the function execution step by step 4. For modulo operations (%), decompose them: compute the division first, then find the remainder - do NOT compute modulo in a single step 5. For ANY arithmetic expression with more than 2 terms (e.g., a * b + c, x + y - z, etc.), break it down step by step and compute intermediate results - do NOT evaluate complex expressions in one shot 6. Calculate the value at position ⟨N⟩ 7. Present your reasoning followed by the final answer

Output Format: JSON object with "reasoning" (detailed step-by-step analysis) and "answer" (predicted number as string) fields.

---

# H ANALYZER OUTPUT EXAMPLES

> **Analysis Output Example**
>
> 1. **Performance Overview:**
>    - Current score: 0.18666 (28% improvement from initial 0.17333)
>    - Inconsistent performance across iterations (scores: 0.17→0.16→0.0→0.22→0.0→0.22)
>    - Best performance achieved in iterations 3 and 5 with score 0.22666
> 2. **Key Error Patterns:**
>    **A. Mathematical Reasoning Failures:**
>    - Complex geometric problem (#70) shows systematic failure to:
>      - Properly handle cyclic quadrilateral properties
>      - Apply collinearity constraints
>      - Use similar triangle relationships effectively
>    - Function intersection problem (#63) reveals:
>      - Inadequate periodic function analysis
>      - Poor handling of composite functions
>      - Failure to systematically count intersection points
>
>    **B. Framework Implementation Issues:**
>    - Overcomplicated input processing attempts (iterations 1-2)
>    - Inconsistent JSON output formatting
> 3. **Successful Elements:**
>    - Basic mathematical structure identification
>    - Geometric component recognition
>    - Clear prompt organization with validation steps
>    - Strong JSON output examples
> 4. **Improvement Pattern Analysis:**
>    **Most Effective Changes (Iterations 3/5):**
>    - Balanced mathematical structure detection
>    - Clear geometric reasoning guidance
>    - Simple but effective input processing
>    - Strong output format examples
>
>    **Counterproductive Changes:**
>    - Overly complex pattern analysis (iteration 1)
>    - Excessive preprocessing (iteration 2)
>    - Oversimplified approach (iteration 4)
> 5. **Recommended Focus Areas:**
>    **A. Input Processing:**
>    - Keep `clean_problem` and `math_structure`
>    - Focus `geometric_analysis` on fundamental properties
>    - Avoid complex pattern detection
>
>    **B. Prompt Engineering:**
>    - Strengthen systematic proof structure
>    - Enhance case analysis guidance
>    - Maintain clear validation steps
>
>    **C. Mathematical Framework:**
>    - Add explicit proof technique selection
>    - Improve periodic function analysis
>    - Strengthen geometric relationship handling
> 6. **Framework-Compatible Implementation:**
>    - Use computed inputs for basic math structure [...]

# I  ANALYZER AND IMPROVER AGENTS: PROMPT TEMPLATES

Table 6: Analyzer and Improver agents. Temperature is set to 0.7 to encourage exploration.

| **Analyzer Agent** |
| --- |
| Instructions: You are the Analyzer agent in the Iterative Task Improvement Pattern that evaluates LLM agent performance on tasks by examining inputs, outputs, and ground truth. Analyze error patterns and suggest framework-compatible improvements.
LLM Agent Framework: ⟨Framework Documentation⟩
You are an analyzer agent that works as part of this iterative improvement workflow: ⟨Iterative Improvement Documentation⟩
Current Task Agent Configuration: ⟨Current Task Config⟩
Current Task Agent Implementation: ⟨Current Task Implementation⟩
Evaluation Summary: ⟨Evaluation Summary⟩
Examples Where Task Agent Was CORRECT: ⟨Correct Examples⟩
Examples Where Task Agent Made ERRORS: ⟨Incorrect Examples⟩
Iteration History: ⟨Iteration History⟩
Analysis Focus: Error patterns across multiple runs, Framework constraints (single LLM call, config-driven), General improvements (avoid overfitting to sample data), Past iteration results and performance trends, Which changes had positive/negative effects on performance, Correlation between specific improvements and score changes
Output Format: ⟨Output Format Requirements⟩ |
| **Improver Agent** |
| Instructions: You are the Improver agent in the Iterative Task Improvement Pattern. Generate improved task agents that follow LLM Agent Framework constraints (single LLM call, config-driven architecture). IMPORTANT: The goal is to create agents that generalize well to UNSEEN data from the same distribution. The current development data is only for evaluation - focus on improvements that will work on new, unseen problems rather than overfitting to the specific development examples.
LLM Agent Framework: ⟨Framework Documentation⟩
Iterative Improvement Pattern: ⟨Iterative Improvement Documentation⟩
Iteration History: ⟨Iteration History⟩
Current Task Agent: ⟨Current Task Agent⟩
Best Performing Agent (Score: ⟨Best Agent Score⟩ from iteration ⟨Best Agent Iteration⟩): ⟨Best Task Agent⟩
Analysis Results: ⟨Analysis Results⟩
Improvement Guidelines: Framework Implementation Rules: All improvements must work within single LLM call architecture, Add computed inputs with processing functions for data transformation, Reference new inputs in prompt templates using {inputs.input_name}, Only implement class methods referenced by computed inputs. Explain changes like: 'Added computed input X that processes Y using function Z, referenced in prompt template as {inputs.X} to achieve improvement W'. IMPORTANT: Remember you can only modify config fields with _modifiable set to True.
Format Requirements: ⟨Output Format Requirements⟩ |

# J AGENT GENERATED BY SONNET IN A PRNG-BBS IMPROVEMENT LOOP

```json
{
  "description": "Enhanced Func1 solver with comprehensive arithmetic validation",
  "input_schema": "func1_solver_input_schema.json",
  "model": {
    "_modifiable": true,
    "name": "anthropic.claude-3-5-sonnet-20241022-v2:0",
    "parameters": {
      "temperature": 0.0,
      "max_tokens": 50000
    }
  },
  "inputs": {
    "_modifiable": true,
    "function_implementation": {
      "source": "computed",
      "function": "extract_implementation",
      "args": {
        "problem_data": "raw_problem_data"
      }
    },
    "max_value_validator": {
      "source": "computed",
      "function": "max_value_validator",
      "args": {
        "problem_data": "raw_problem_data"
      }
    },
    "value_tracker": {
      "source": "computed",
      "function": "value_tracker",
      "args": {
        "problem_data": "raw_problem_data"
      }
    },
    "checksum_generator": {
      "source": "computed",
      "function": "checksum_generator",
      "args": {
        "problem_data": "raw_problem_data"
      }
    },
    "division_validator": {
      "source": "computed",
      "function": "division_validator",
      "args": {
        "problem_data": "raw_problem_data"
      }
    },
    "quotient_checker": {
      "source": "computed",
      "function": "quotient_checker",
      "args": {
        "problem_data": "raw_problem_data"
      }
    },
    "range_validator": {
      "source": "computed",
      "function": "range_validator",
      "args": {
        "problem_data": "raw_problem_data"
      }
    },
    "function_steps": {
      "source": "computed",
      "function": "extract_function_steps",
      "args": {
        "problem_data": "raw_problem_data"
      }
    },
    "hyperparams": {
      "source": "computed",
      "function": "extract_hyperparams",
      "args": {
        "problem_data": "raw_problem_data"
      }
    },
    "initial_value": {
      "source": "computed",
```

```
 79          "function": "extract_initial_value",
 80          "args": {
 81            "problem_data": "raw_problem_data"
 82          }
 83        },
 84        "n": {
 85          "source": "computed",
 86          "function": "get_n_parameter",
 87          "args": {
 88            "n": 2
 89          }
 90        },
 91        "raw_problem_data": {
 92          "source": "data",
 93          "path": "$",
 94          "required": true
 95        }
 96      },
 97      "prompt": {
 98        "_modifiable": true,
 99        "sections": {
100          "instructions": "You are a precise arithmetic calculator focused on careful validation
                 of each calculation step. Follow ALL validation requirements exactly and show every
                 intermediate value and check.",
101          "task_description": "Calculate the value at position {inputs.n} using rigorous
                 validation. Start with initial_value={inputs.initial_value}. Record and verify ALL
                 intermediate values.",
102          "validation_requirements": "Large Number Validation:\n{inputs.max_value_validator}\n\
                 nValue Chain Tracking:\n{inputs.value_tracker}\n\nChecksum Validation:\n{inputs.
                 checksum_generator}\n\nDivision Validation:\n{inputs.division_validator}\n\
                 nQuotient Checking:\n{inputs.quotient_checker}\n\nRange Validation:\n{inputs.
                 range_validator}",
103          "function_steps": "Calculation Steps:\n{inputs.function_steps}",
104          "output_format": "Respond with JSON:\n{\n  \"reasoning\": \"Show each arithmetic step
                 with ALL intermediate values, validation checks, checksums, and cross-validations
                 \",\n  \"answer\": \"final numeric result as string\"\n}"
105        },
106        "template": "### Instructions:\n{prompt.sections.instructions}\n\n### Task:\n{prompt.
                 sections.task_description}\n\n### Implementation:\n{inputs.function_implementation}\n
                 \n### Hyperparameters:\n{inputs.hyperparams}\n\n### Required Validation Steps:\n{
                 prompt.sections.validation_requirements}\n\n### Calculation Process:\n{prompt.
                 sections.function_steps}\n\n### Output Format:\n{prompt.sections.output_format}\n\
                 nCalculate step by step, showing all validation:"
107      },
108      "output": {
109        "_modifiable": false,
110        "format": "json",
111        "schema": {
112          "type": "object",
113          "required": [
114            "reasoning",
115            "answer"
116          ],
117          "properties": {
118            "reasoning": {
119              "type": "string"
120            },
121            "answer": {
122              "type": "string"
123            }
124          }
125        },
126        "error_values": {
127          "reasoning": "Error occurred during reasoning.",
128          "answer": "PARSE_ERROR"
129        }
130      },
131      "settings": {
132        "_modifiable": true,
133        "class_name": "Func1SolverAgent",
134        "timeout_seconds": 300
135      }
136 }
```

```python
1  #!/usr/bin/env python3
2  """Enhanced Func1 solver with comprehensive arithmetic validation"""
3
4  import sys
5  import os
6
```

```python
sys.path.append(os.path.join(os.path.dirname(__file__), '..', '..', '..', '..', 'llm_agent'))

from llm_agent import LLMAgent

class Func1SolverAgent(LLMAgent):
    """Enhanced Func1 solver with arithmetic validation."""

    def extract_implementation(self, problem_data):
        """Extract function implementation from problem data."""
        return problem_data.get('implementation', '')

    def extract_hyperparams(self, problem_data):
        """Extract and format hyperparameters."""
        hyperparams = problem_data.get('hyperparams', {})
        if isinstance(hyperparams, dict):
            formatted = []
            for key, value in hyperparams.items():
                formatted.append(f"{key}: {value}")
            return "\n".join(formatted)
        return str(hyperparams)

    def extract_initial_value(self, problem_data):
        """Extract initial value from problem data."""
        return str(problem_data.get('initial_value', ''))

    def get_n_parameter(self, n):
        """Return the position parameter."""
        return str(n)

    def max_value_validator(self, problem_data):
        """Generate validation steps for large number arithmetic."""
        steps = [
            "Large Number Validation:",
            "1. For multiplication A     B:",
            "   a. Split into single digits: A =   a a ... a , B =   b b ... b ",
            "   b. Calculate partial products:",
            "      - For each digit pair ( a , b ):   p     = a         b ",
            "      - Track carries explicitly",
            "   c. Sum partial products with position tracking",
            "   d. Verify: result = sum of all partial products",
            "2. Cross-validate using reverse calculation:",
            "   - Split result into chunks",
            "   - Verify chunks match partial products"
        ]
        return "\n".join(steps)

    def value_tracker(self, problem_data):
        """Generate value tracking steps for calculation chain integrity."""
        steps = [
            "Value Chain Tracking:",
            "1. For each position i in calculation:",
            "   a. Record value[i] = current value",
            "   b. Record squared[i] = value[i]  ",
            "   c. Record quotient[i] = squared[i]     modulus",
            "   d. Record remainder[i] = value[i+1]",
            "2. Verify chain integrity:",
            "   - remainder[i] = squared[i] - (quotient[i]     modulus)",
            "   - 0        remainder[i] < modulus",
            "   - value[i+1] = remainder[i]"
        ]
        return "\n".join(steps)

    def checksum_generator(self, problem_data):
        """Generate checksum validation steps between calculations."""
        steps = [
            "Checksum Validation:",
            "1. For each step i:",
            "   a. Calculate checksum[i] = (value[i]     31 + squared[i]) mod 997",
            "   b. Verify: checksum[i] matches independent calculation",
            "2. Cross-step validation:",
            "   - Verify: checksum[i+1] = (remainder[i]     31) mod 997",
            "   - Track checksum chain for each position"
        ]
        return "\n".join(steps)

    def division_validator(self, problem_data):
        """Generate enhanced division validation steps."""
        modulus = problem_data.get('hyperparams', {}).get('modulus', 0)
        steps = [
            f"Division Validation Steps:",
            f"1. For dividing number A by {modulus}:",
```

```python
                    f"   a. Calculate quotient Q = A   {modulus}",
                    f"   b. Verify: Q    {modulus}    A",
                    f"   c. Verify: (Q + 1)    {modulus} > A",
                    f"   d. Record Q for remainder calculation",
                    f"   e. Track sub-steps:",
                    f"      – Record partial quotients",
                    f"      – Verify digit-by-digit multiplication",
                    f"      – Cross-validate remainder"
                ]
        return "\n".join(steps)

    def quotient_checker(self, problem_data):
        """Generate enhanced quotient checking steps."""
        steps = [
            "Quotient Verification:",
            "1. Record squared value (S)",
            "2. Calculate and verify quotient (Q):",
            "   – Q = S    modulus (integer division)",
            "   – Verify: Q    modulus    S",
            "   – Verify: (Q + 1)    modulus > S",
            "   – Perform digit-by-digit multiplication check",
            "3. Calculate remainder:",
            "   R = S – (Q    modulus)",
            "4. Verify remainder is correct:",
            "   – 0    R < modulus",
            "   – Cross-validate with value chain"
        ]
        return "\n".join(steps)

    def range_validator(self, problem_data):
        """Generate enhanced range validation steps."""
        modulus = problem_data.get('hyperparams', {}).get('modulus', 0)
        steps = [
            f"Range Validation:",
            f"1. For each position i:",
            f"   – Verify value[i]    0",
            f"   – Verify value[i] < {modulus}",
            f"2. For squared values at position i:",
            f"   – Record exact value squared[i]",
            f"   – Verify squared[i] = value[i]    value[i]",
            f"   – Verify using digit-by-digit multiplication",
            f"3. Position-specific bounds:",
            f"   – Track maximum possible value at each position",
            f"   – Verify values stay within position bounds"
        ]
        return "\n".join(steps)

    def extract_function_steps(self, problem_data):
        """Extract and format key arithmetic operations."""
        steps = [
            "1. Initialize calculation chain:",
            "   – Set value[0] = initial_value",
            "   – Calculate checksum[0]",
            "2. For each position i:",
            "   a. Square Operation:",
            "      – Calculate value[i]    using digit-by-digit multiplication",
            "      – Record squared[i] and partial products",
            "      – Verify checksum[i]",
            "   b. Division and Modulo:",
            "      – Calculate quotient[i] with validation",
            "      – Calculate remainder[i] = value[i+1]",
            "      – Verify chain integrity",
            "   c. Cross-validate:",
            "      – Verify all checksums match",
            "      – Confirm value chain consistency",
            "      – Validate position-specific bounds"
        ]
        return "\n".join(steps)
```

# K    ACCURACY-DRIVEN IMPROVEMENT LOOPS

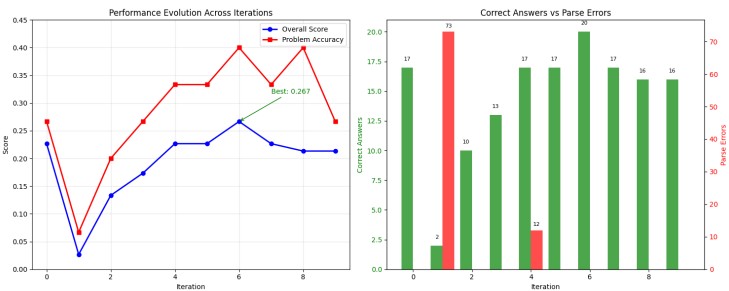

Figure 2: Aime 2024 improvement loop using Sonnet 3.5 v2 as meta agents.

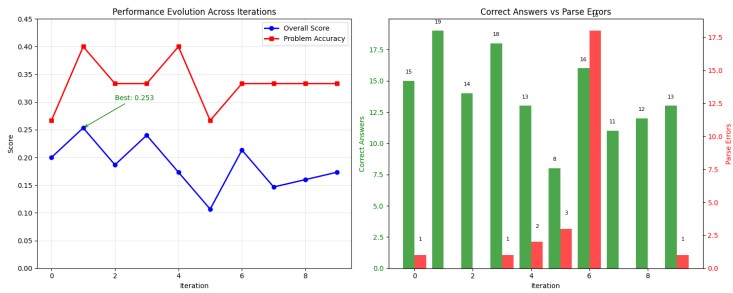

Figure 3: Aime 2024 improvement loop using OpenAI 120b reasoning model as meta agents.

## L  AGENT GENERATED BY ACCURACY+COST OPTIMIZATION LOOP BY GPT-OSS-120B

```
1  "description": "AIME 2024 ICL task agent \u2013 focused rhombus example, explicit bound,
       streamlined prompt",
2    "input_schema": "aime2024_icl_solver_input_schema.json",
3    "model": {
4      "_modifiable": true,
5      "name": "anthropic.claude-3-5-sonnet-20241022-v2:0",
6      "parameters": {
7        "temperature": 0.2,
8        "max_tokens": 500
9      }
10   },
11   "inputs": {
12     "_modifiable": true,
13     "problem_text": {
14       "source": "computed",
15       "function": "extract_problem_text",
16       "args": {
17         "problem_data": "raw_problem_data"
18       }
19     },
20     "rhombus_property": {
21       "source": "computed",
22       "function": "rhombus_property",
23       "args": {}
24     },
25     "solve_perpendicular_condition": {
26       "source": "computed",
27       "function": "solve_perpendicular_condition",
28       "args": {
29         "problem_data": "raw_problem_data"
30       }
31     },
32     "bd2_bound": {
33       "source": "computed",
34       "function": "bd2_bound",
35       "args": {}
36     },
37     "rhombus_icl_example": {
38       "source": "computed",
39       "function": "rhombus_icl_example",
40       "args": {}
41     },
42     "guard_text": {
43       "source": "computed",
44       "function": "guard_text",
45       "args": {}
46     },
47     "raw_problem_data": {
48       "source": "data",
49       "path": "$",
50       "required": true
51     }
52   },
53   "prompt": {
54     "_modifiable": true,
55     "sections": {
56       "instructions": "You are a mathematical problem solver. Solve the given AIME problem and
               output the answer as a three\u2011digit integer.",
57       "output_format": "You must respond with a JSON object containing exactly two fields:\n-
               \"reasoning\": string with your detailed solution\n- \"answer\": string with a
               three\u2011digit answer (pad with leading zeros if needed)\n\nFormat: {\"reasoning
               \": \"Your detailed solution here\", \"answer\": \"XXX\"}\n\nExample:\n{\"reasoning
               \": \"To solve..., I ...\", \"answer\": \"042\"}"
58     },
59     "template": "### Instructions:\n{prompt.sections.instructions}\n\n### Geometry hint:\n{
           inputs.rhombus_property}\n\n### Algebraic bound hint:\n{inputs.
           solve_perpendicular_condition}\n\n### Known bound value:\nThe minimal possible BD\
           u00b2 is {inputs.bd2_bound}.\n\n### Example to follow:\n{inputs.rhombus_icl_example}\
           n\n### Problem:\n{inputs.problem_text}\n\n### Output format:\n{prompt.sections.
           output_format}\n\n{inputs.guard_text}"
60   },
61   "output": {
62     "_modifiable": false,
63     "format": "json",
64     "schema": {
65       "type": "object",
```

```
 66        "required": [
 67          "reasoning",
 68          "answer"
 69        ],
 70        "properties": {
 71          "reasoning": {
 72            "type": "string"
 73          },
 74          "answer": {
 75            "type": "string",
 76            "pattern": "^[0-9]{3}$"
 77          }
 78        }
 79      },
 80      "error_values": {
 81        "reasoning": "Error occurred during reasoning.",
 82        "answer": "PARSE_ERROR"
 83      }
 84    },
 85    "settings": {
 86      "_modifiable": false,
 87      "class_name": "AIME2024ICLSolverAgent",
 88      "timeout_seconds": 300,
 89      "exit_on_parse_failure": false
 90    }
 91 }
```

```python
 1  from llm_agent import LLMAgent
 2  from typing import Dict, Any
 3
 4  class AIME2024TaskAgent(LLMAgent):
 5      """
 6      AIME2024-specific reasoning + answer agent.
 7
 8      Processes AIME mathematical competition problems and generates detailed step-by-step
 9          reasoning followed by answers.
    """
10
11      def __init__(self, name="AIME2024_Task", config_path=None, dry_run=False):
12          super().__init__(name, config_path, dry_run)
13
14      def extract_problem_text(self, problem_data: Dict[str, Any]) -> str:
15          """
16          Extract problem text from AIME2024 data format.
17
18          AIME2024 format:
19          {
20              "id": int,
21              "problem": str,  # Main problem statement
22              "url": str
23          }
24          """
25          return str(problem_data.get('problem', ''))
```

