# OpenReview forum: "Self-Directed Discovery: How LLMs Explore and Optimize Configurable LLM Solutions"
_ICLR.cc/2026/Conference — ICLR 2026 Conference Desk Rejected Submission_

### Official Review · Reviewer_stmz · 2025-10-31

**Soundness:** 2
**Presentation:** 2
**Contribution:** 2
**Rating:** 2
**Confidence:** 3

**Summary:**

This paper proposes a configuration-driven framework that enables off-the-shelf large language models (LLMs) to automatically create and iteratively improve LLM-based workflows. The core idea is to formalize LLM agents via declarative configurations (JSON + Python + schema files) and use meta-agents (Analyzer and Improver) to optimize these agents through iterative self-improvement loops. Experiments on reasoning-heavy benchmarks such as AIME, Math500, MMLU-Pro, and synthetic PRNG tasks demonstrate that such self-directed improvement loops can yield better-performing and more interpretable solutions compared to baseline prompt-engineering methods.

**Strengths:**

Clear System Design:
The configuration-based agent definition is well-structured, modular, and interpretable. The separation of configuration, code, and schema files provides a principled way to balance flexibility and transparency.

Self-Referential Framework:
Defining both task agents and meta-agents (Analyzer, Improver) within the same configuration system is an elegant concept. It enables recursive self-optimization without external meta-prompt engineering.

Comprehensive Experiments:
The paper evaluates across diverse tasks—mathematical reasoning (AIME, Math500), multi-domain QA (MMLU-Pro), and algorithmic sequence prediction (PRNG). The setup illustrates the framework’s generality.

Artifact-driven Optimization:
The idea of using documentation and prior agent examples instead of meta-prompts is conceptually novel and closer to a self-evolving agent knowledge base.

**Weaknesses:**

Limited Novelty vs Existing Frameworks:
While the configuration-driven approach is nicely implemented, the conceptual novelty is somewhat incremental compared to prior works such as DSPy, TextGrad, GPTSwarm, and PromptBreeder. The paper mainly integrates existing ideas (prompt optimization, agent frameworks, LLM self-improvement) into a unified system without introducing a fundamentally new algorithmic principle.

Evaluation Depth:

Improvements on AIME and MMLU-Pro are modest (e.g., +0.02–0.05 absolute gains).

The PRNG experiments, while interesting, rely on artificially simplified setups and may not convincingly demonstrate real-world generalization.

No ablation or comparison with existing LLM optimizer frameworks (e.g., AutoGPT, AIDE, DSPy) is provided.

Scalability and Efficiency:
The experiments rely heavily on Claude 3.5 Sonnet and GPT-OSS-120B, but runtime cost, scalability, and reproducibility details are missing. The claim of “self-improving” remains qualitative rather than quantitatively substantiated across longer iterations or larger datasets.

Overstated Claims:
The paper suggests bridging “generic code-generation” and “prompt engineering” with self-evolving knowledge accumulation, yet the evidence for cumulative improvement or cross-task transfer is not shown.

Writing and Positioning:
The exposition is verbose and occasionally repetitive. The distinction between agent generation and improvement phases could be clarified with more concrete algorithmic pseudocode or diagrams.

**Questions:**

N/A

---

### Official Review · Reviewer_dcYb · 2025-10-31

**Soundness:** 3
**Presentation:** 2
**Contribution:** 2
**Rating:** 2
**Confidence:** 3

**Summary:**

This paper proposes a configuration-driven framework for automatically building and iteratively improving LLM-based solutions. The framework defines agents through JSON configurations, Python implementations, and input schemas, enabling LLMs to autonomously generate and optimize workflows.

**Strengths:**

1. The design of iterations based on Analyzer and Improver is technically sounded.
2. The appendix provides numerous details, which help enhance reproducibility.

**Weaknesses:**

1. The datasets used in the experiments are primarily related to mathematics, which raises concerns about the broad applicability of the method. It is recommended to conduct experimental evaluations on more diverse datasets.
2. The innovation and significance of this manuscript are relatively lacking. Specifically, it is difficult to understand from the current content where the challenges of designing such a system lie, what shortcomings of existing methods it addresses, and what innovative contributions it makes. Overall, this manuscript reads more like a technical report than an academic paper.
3. As seen in Figure 2, this improvement loop is not monotonic but rather an exploratory process. This indicates that the system has a high degree of randomness, instead of a reliable optimizer that can steadily converge to an optimal solution.
4. This work lacks a detailed comparison with existing methods in terms of computational cost, making it difficult to assess the practical value of the proposed approach.

**Questions:**

What contributions does this manuscript provide that could inspire other readers?

---

### Official Review · Reviewer_37Kv · 2025-11-03

**Soundness:** 2
**Presentation:** 2
**Contribution:** 2
**Rating:** 2
**Confidence:** 3

**Summary:**

The paper proposes a self-improving framework where LLMs are used to both create and iteratively optimize other LLM-based solutions (termed "agents").

**Strengths:**

The core idea of using a structured, configuration-driven approach to make LLM workflows amenable to automated optimization is timely and compelling.

**Weaknesses:**

However, the claims in this work is overstated. The paper uses ambitious terms like "self-directed discovery" and "self-evolving system", but the implemented mechanism is a relatively simple, fixed iterative loop. The framework itself is a neat piece of engineering, but its contributions to the fundamental machine learning or optimization are limited. Plus, there are some critical flaws in the methods and evaluation.

**Questions:**

1. Your framework defines an "LLM Agent" as a workflow making a single, isolated LLM inference call (Lines 079-080). This seems to be a significant oversimplification. How does this "self-directed discovery" scale beyond these trivial 'prompt-and-parse' tasks to the complex, multi-step reasoning chains, tool use, and memory management that characterize the truly powerful and challenging agentic systems currently being explored by the research community?

2. The paper claims a "documentation-driven" approach with "minimal generic prompts" for the meta-agents (Analyzer and Improver), suggesting a low degree of manual effort (Lines 066-071). However, the success of your meta-agents does not depend on the prompts alone, but on a rich, highly-structured context including the entire framework's documentation, curated examples, and detailed evaluator outputs. Isn't this simply shifting the burden of "prompt engineering" to "documentation and context engineering"? How can you claim the system is "minimal" when its success is critically dependent on a large and meticulously structured set of contextual artifacts that are themselves products of significant expert design?

3. The term "self-improving system" is used throughout the paper, implying a system that can enhance its own core capabilities over time. Yet, the core optimization engine—the Analyzer and Improver agents—remains static and is not itself the subject of optimization within the experimental loop. Given that these meta-agents are the most critical component, and they are not themselves improved, in what meaningful sense is the system itself self-improving, rather than merely executing a fixed, pre-programmed search heuristic on a separate, subordinate task?

4. The improvement loop is presented as a central contribution, but it appears to be a form of greedy, local search guided by LLM-generated proposals. The evaluation compares the performance of the final agent to the initial agent but fails to compare the optimization process itself against any established baselines. Why are there no comparisons to standard black-box optimization techniques, such as Bayesian optimization, genetic algorithms, or even a simple random search over the configuration space, to benchmark the efficiency and efficacy of your LLM-driven loop?

---

### Official Review · Reviewer_eyuk · 2025-11-03

**Soundness:** 2
**Presentation:** 2
**Contribution:** 2
**Rating:** 2
**Confidence:** 4

**Summary:**

The paper proposes a framework for automated configuration of LLM invocation via a constraint exploratory system. The invocation is parametrized by prompt templates, input transformations and model parameters. Each of these are tunable in an iterative process through an analyzer and improver agent. The method is evaluated on MMLU-pro, AIME, and a novel task of prediction pseudo-random-numbers in sequence.

**Strengths:**

- The paper addresses the problem of defining task-specific llm invocation mechanisms.
- The paper defines a clear parametrization for llm-invokation via templates and input transformations.
- The paper aims at a human-interpretable workflow that can be easily interpreted and understood, as opposed to a black-box solution.

**Weaknesses:**

# Main weaknesses
- The paper talks about agents, but the definitions of "agent" is incompatible with the common use of the word in the AI/LLM community. Usually agents invoke LLMs iteratively, usually with a variable number of calls that is determined by the LLM. This meaning of the word is used in the "AgentSDK". This agrees with the definition of Agent on wikipedia and the use of Agent as defined by many cloud an LLM providers. However, this paper defines an agent to contain a single LLM invocation, which to me is missing the fundamental criteria of an Agent.

- The paper proposes a new framework for automatic tuning for specific tasks. As mentioned in the references, prompt-tuning and self-improvement are common research topics right now. However, there is no comparison made with any method from the literature, and there is no ablation study about whether the proposed decomposition of the problem into prompt templates, configuration and input transformations is beneficial.

- The paper is missing some critical references, in particular the Darwin-Goedel machine, AlphaEvolve and "An AI system to help scientists write expert-level empirical software", all of which involve improving systems that are actually agentic in nature.

# Other issues
- The summary claims that the proposed systems is human-understandable, unlike other solutions. However, there is no discussion or evidence of this in the paper.

- The paper lists Claude 3.5 as a non-reasoning model. However, the model sometimes does produce thinking tokens, and it's somewhat unclear how to define reasoning models, and how to categorize existing models, given that their training is not public. In particular, the mention of test-time scaling clearly points towards this being a reasoning models as non-reasoning models don't exhibit test-time scaling.


# Minor comments
Line 113: "modifiable" should be using textit, not math.

Regarding the PRNG task, the model might still recognize the algorithms given the python code. The motivation and purpose for this task seem quite unclear to me.

Line 430 is missing parenthesis for the APE citation

Line 450 is missing parenthesis for the React citation.

Appendix G: k should be called n given the definitions above.

**Questions:**

- Can you provide a reference for Agents being defined using only one llm call?

- Can you compare your approach against related methods, in particular alpha-evolve-type systems and prompt tuning systems?

---

### Official Review · Reviewer_YuvY · 2025-11-08

**Soundness:** 3
**Presentation:** 3
**Contribution:** 3
**Rating:** 6
**Confidence:** 3

**Summary:**

This paper explores the use of powerful off-the-shelf LLMs to automatically construct and iteratively improve LLM-based solutions. The authors propose a configuration-driven framework that formalizes workflows through model parameters, prompt templates, and data transformations, enabling self-referential optimization loops defined within the same structure.

**Strengths:**

1. Experiments on challenging datasets demonstrate that the approach can autonomously discover improved, interpretable, and verifiable solutions.

2. Overall, the work presents a minimal yet generic self-improving system that connects general-purpose automatic optimization with domain-specific prompt engineering techniques.

3. Presentations are clear with abundant case studies.

**Weaknesses:**

1. It is unclear how the proposed approach can be extended or adapted to multi-agent systems. The authors are encouraged to discuss the potential challenges and feasibility of such a transfer.

2. It is recommended that the Related Work section be placed immediately after the Introduction to help readers more easily follow the background and motivation of the study.

3. The paper would benefit from a clear diagram illustrating the overall pipeline of the proposed framework. Such a visual overview is essential for improving the clarity, accessibility, and reproducibility of the work.

**Questions:**

See weaknesses.

---

### Note · Program_Chairs · 2026-01-17
**Submission Desk Rejected by Program Chairs**

The following references in this submission do not refer to real documents and/or have major errors in bibliographic information:

 Tianyi Huang, Huimin Yao, Jiacheng He, Zhiyi Zheng, and Zhou Lin. TextGrad: Autodifferentiating through arbitrary text with large language models. arXiv preprint arXiv:2308.02450, 2023.